# The *Helicobacter pylori* AI-clinician harnesses artificial intelligence to personalise *H. pylori* treatment recommendations

Kyle Higgins [1,20], Olga P. Nyssen [2,20], Joshua Southern[3], Ivan Laponogov[1], AIDA CONSORTIUM*, Dennis Veselkov[1,3], Javier P. Gisbert [2,21] ✉, Tania Fleitas Kanonnikoff[4,5,21] ✉ & Kirill Veselkov [1,6,21] ✉

*Helicobacter pylori* (*H. pylori*) is the most common carcinogenic pathogen globally and the leading cause of gastric cancer. Here, we develop a reinforcement learning-based AI Clinician system to personalise treatment selection and evaluate its ability to improve eradication success compared to clinician-prescribed therapies. The model is trained and internally validated on 38,049 patients from the retrospective European Registry on *Helicobacter pylori* Management (Hp-EuReg), using independent state deep Q-learning (isDQN) to recommend optimal therapies based on patient characteristics such as age, sex, antibiotic allergies, country, and pre-treatment indication. In internal validation using real-world Hp-EuReg data, AI-recommended therapies achieve a 94.1% success rate (95% CI: 93.2–95.0%) versus 88.1% (95% CI: 87.7–88.4%) for clinician-prescribed therapies not aligned with AI suggestions—an improvement of 6.0%. Results are replicated in an external validation cohort (*n* = 7186), confirming generalisability. The AI system identifies optimal treatment strategies in key subgroups: 65% (*n* = 24,923) are recommended bismuth-based therapies, and 15% (*n* = 5898) non-bismuth quadruple therapies. Random forest modelling identifies region and concurrent medications as patient-specific drivers of AI recommendations. With nearly half the global population likely to contract *H. pylori*, this approach lays the foundation for future prospective clinical validation and shows the potential of AI to support clinical decision-making, enhance outcomes, and reduce gastric cancer burden.

*Helicobacter pylori (H. pylori)* is a Gram-negative S-shaped bacteria which has adapted to colonize the niche of the deep gastric mucous layer in the human stomach[1]. *H. pylori* infection in the gastric mucosa leads to a diverse inflammatory response in local epithelial cells, resulting in chronic active gastritis. (Fig. 1a) Despite producing antibodies to *H. pylori* antigens, this immune response is generally incapable of eradicating the bacteria. Over decades, this inflammation has been thought to lead to a variety of conditions, most notably

[1]Division of Cancer, Department of Surgery and Cancer, Faculty of Medicine, Imperial College London, London, UK. [2]Gastroenterology Unit, Hospital Universitario de La Princesa, Instituto de Investigación Sanitaria Princesa (IIS-Princesa), Universidad Autónoma de Madrid (UAM), Centro de Investigación Biomédica en Red de Enfermedades Hepáticas y Digestivas (CIBEREHD), Madrid, Spain. [3]Department of Computing, Faculty of Engineering, Imperial College London, London, UK. [4]Instituto Investigación Sanitaria INCLIVA (INCLIVA), Medical Oncology Department, Hospital Clínico Universitario de Valencia, Valencia, Spain. [5]Centro de Investigación Biomédica en Red Cáncer (CIBERONC), Instituto de Salud Carlos III, Madrid, Spain. [6]Department of Environmental Health Sciences, Yale University, New Haven, CT, USA. [20]These authors contributed equally: Kyle Higgins, Olga P. Nyssen. [21]These authors jointly supervised this work: Javier P. Gisbert, Tania Fleitas Kanonnikoff, Kirill Veselkov.*A list of authors and their affiliations appears at the end of the paper. ✉e-mail: javier.p.gisbert@gmail.com; tfleitas@incliva.es; kirill.veselkov04@imperial.ac.uk

**Fig. 1 | Hp-EuReg dataset and AI clinician overview. a** *Helicobacter pylori* (*H. pylori*) infects the stomach of around one in two individuals worldwide. It does so by infiltrating the gastric mucosa, aided by a highly motile flagellum. The infection is characterized by an overall increase in the acidity of the gastric fluid, onset by urease production, and increased inflammation in the gastric epithelia, often spanning decades before diagnosis. **b** *H. pylori*-induced pathology most commonly includes gastric cancer (at least one per one hundred infected individuals) and peptic ulcer disease (around one in ten infected individuals). **c** The Hp-EuReg project is an international, multicenter prospective registry collecting *H. pylori* treatment management strategies across Europe and including to date over 75,000 patient records. **d** The data in this registry includes patient metadata, treatment strategy employed by the clinician, and result of this treatment, in terms of eradication. **e** The *H. pylori* AI-clinician is trained on the Hp-EuReg dataset and designed to provide patient-specific optimal treatment recommendations for *H. pylori* eradication. Created in BioRender. Higgins, K. (2025) https://BioRender.com/qi85m5d.

peptic ulcer disease (PUD) and gastric cancer. Of the nearly 4 billion people infected by *H. pylori*, ~10% will develop PUD within a decade of infection, meaning roughly 780 million worldwide will be afflicted by this condition[2]. (Fig. 1b) Peptic ulcers result from damage to the lining of the stomach and may lead to complications such as internal bleeding and perforations, with a high mortality rate in such cases. *H. pylori* eradication has shown promising results in the treatment of PUD, achieving not only ulcer healing but also preventing its recurrence.

Around 90% of gastric cancer cases are due to *H. pylori* infection[3]. It is estimated that gastric cancer makes up 37% of chronic infection-induced cancers, making *H. pylori* the most frequent carcinogenic pathogen[4]. Gastric cancer is thought to develop after years of inflammation-induced gastric atrophy, wherein achlorhydria drives the development of an abnormal microbiome, further driving the transformation of gastric epithelial cells to an oncogenic state (a hypothesis termed the 'Correa cascade')[5]. Indeed, a 'point of no return' has been observed with regard to *H. pylori* infection in patients developing gastric cancer, past which *H. pylori* eradication is insufficient to interrupt the inflammatory cascade leading to oncogenesis. Despite advancements in treatment such as chemotherapy and surgery, gastric cancer results in a poor prognosis compared to other cancer types, especially in advanced stages[6].

The European Registry on *Helicobacter pylori* management (Hp-EuReg) was established to combat the high social and health burden of *H. pylori* infection across Europe. It was noted at the time that consensus and clinical guidelines were established for *H. pylori* treatment, but that no data existed cataloguing the implementation of these recommendations[7]. This project took the form of an international and multicenter prospective non-interventional registry documenting the real clinical practice by European gastroenterologists of *H. pylori* management in the majority of countries across Europe. (Fig. 1c) Patient data documented include several demographics categories (i.e., country, sex, age), pre-existing gastrointestinal symptoms, treatment indication, previous eradication attempts, and compliance. (Fig. 1d) Crucially, this registry documents treatment chosen, duration of treatment, proton pump inhibitor (PPI) dosage, and eradication outcome. To date, this registry has been used in over 40 published studies[8]. (For a full list of publications, see ref. 9) The most common uses for this dataset have been to assess treatment effectiveness, especially in a country or region-specific context. However, most of these studies have mainly relied on traditional statistical methods rather than advanced methods employing machine learning (ML) and artificial intelligence (AI), with one notable exception[10].

The most frequently used treatments in this registry include the administration of triple and quadruple (either bismuth or non-bismuth based) antibiotics regimens. Treatment durations predominantly include 7, 10, and 14-day prescriptions. Components of these treatments include a combination of at least two antibiotics and a proton pump inhibitor (PPI) in order to raise stomach pH and bismuth for its bacteriostatic effect. Standard triple therapies, most often consisting of two antibiotics (amoxicillin and clarithromycin) and a PPI, were a great advance in the treatment of *H. pylori* in the 1990s, leading to its previous adoption as the treatment gold standard[11]. However, increasing clarithromycin resistance (up to 23% observed in Hp-EuReg)[8] and other factors have led to the development of additional therapies such as quadruple regimens. Certain formulations of quadruple therapies quickly demonstrated >90% eradication rate (now considered the threshold for an optimal *H. pylori* regimen[12]), leading to their adoption as the current recommendation standard[13]. In their traditional formulation, they combine a nitroimidazole (such as metronidazole or tinidazole) with a PPI and antibiotics amoxicillin, clarithromycin. However, due in part to an antibiotic resistance, an alternative bismuth quadruple regimen has been widely adopted (including tetracycline and metronidazole), to great effect in first-line treatments[14,15]. Bismuth has been included for its bactericidal effect, rendering bismuth quadruple therapy unaffected by clarithromycin and metronidazole resistance[16]. Sequential therapies were also developed in part to overcome limitations posed by triple therapies and consist of a two-part treatment period, first using a PPI and amoxicillin, followed by a PPI, clarithromycin, and either tinidazole or metronidazole[17]. However, sequential therapies have been administered to variable effect, with eradication rates varying from 80% to 90%, largely dependent on region[18–20]. Finally, bismuth single capsule therapies such as Pylera® replace multi-drug regimens with a single pill containing bismuth, metronidazole, and tetracycline, combined with a PPI[21]. Such therapies are relatively new, with Pylera® first approved by the FDA in 2006 and currently only approved in Europe in a subset of countries[22]. Early studies show the eradication rate varying from 80% to around 95%[23,24], and a recent meta-analysis report an effective eradication (90%) not only in the first-line but also in rescue therapy and in those patients with clarithromycin- or metronidazole-resistant strains, and in those previously treated with clarithromycin[25]. Though further study is needed as implementation is increased in diverse populations.

A further consideration for treatment recommendation is the presence of allergies to penicillin-like medications, such as amoxicillin. Around 1–5% of patients globally have documented penicillin allergies[26], though higher percentages suggested when self-reporting[27]. The presence of this allergy necessitates use of therapies without amoxicillin, such as levofloxacin-based regimens or a tetracycline, metronidazole, and bismuth salts regimen combined with a PPI such as those found in bismuth single capsule therapies.

Despite the richness of the Hp-EuReg dataset, most analyses have relied on conventional statistical methods. Machine learning (ML) and artificial intelligence (AI) methods offer opportunities to capture complex interactions among variables and patterns from the data[10], especially as applied to personalized medicine[28]. Broadly, ML approaches can be categorized into unsupervised, supervised, and reinforcement learning paradigms. Supervised learning relies on labelled data, making it particularly useful for tasks such as label prediction by mapping input features to known outputs. Thus, it is particularly valuable for tasks such as disease classification, patient outcome prediction, and biomarker discovery. Among traditional models, support vector machines (SVMs) are particularly effective in situations where there is a large number of high-dimensional inputs, such as medical imaging tasks and genomic data classification, due to the ability to construct optimal decision boundaries via kernel methods[29]. Similarly, ensemble learning techniques such as Random Forest (RF) improve robustness of modelling by aggregating multiple decision trees, making it well-suited for handling noisy and imbalanced clinical datasets, predicting disease risks, and identifying complex interactions of biomarkers[30,31].

In recent years, these methods have been complemented by deep learning approaches, which apply neural networks to similar tasks. A commonly used example is convolutional neural nets (CNNs) which benefit from abstracting features away from their spatial localization are particularly suited for medical image tasks[32,33]. Transformers were also developed for natural language processing (NLP), excel at capturing long-range dependencies in sequential data, and have recently been used to analyze electronic health records, genomic, and proteomic sequences[34]. Autoencoders, including variational auto-encoders (VAEs)[35,36] have been instrumental for anomaly detection and dimensionality reduction, proving effective for tasks like identifying rare disease features and generating synthetic patient data.

Reinforcement learning (RL) represents a distinct and increasingly significant paradigm, particularly for patient decision-making in medicine. Reinforcement learning does not rely on specific predefined labels which it needs to predict (such as in case of classical supervised machine learning) but rather is trying to maximize the overall reward it can achieve through its actions. This is performed through training the so called virtual agent, which learns how to interact with its environment via trial-and-error to achieve a maximal reward and minimal penalty[28,37]. This framework has seen significant improvements after being refined in fields such as robotics, gaming, and autonomous systems. However, its rich potential has been underutilized in the medicine, despite its strong ability to learn iteratively in dynamic patient care environments.

In this work, we develop the *H. pylori* AI-Clinician which applies RL to provide patient-specific first-line treatment recommendations and determine if personalized treatments would improve eradication rate compared to a single recommended treatment. (Fig. 1e) This method applies RL, which learns iteratively which actions to take (termed policy) to maximize reward in the context of a given state. RL is well-suited for datasets such as Hp-EuReg with many interacting variables (on the order of hundreds to thousands) as it is sensitive to small differences in rewards, and able to detect subtle factors which affect outcome in the state, and therefore patient outcomes[38].

## Results

Using Hp-EuReg, we evaluated the consistency of the AI Clinician with different splits of training data by generating 500 independent models by ten-fold, fifty repeat cross validation. For each repeat, training (model optimization) was performed using a 90% random sample of

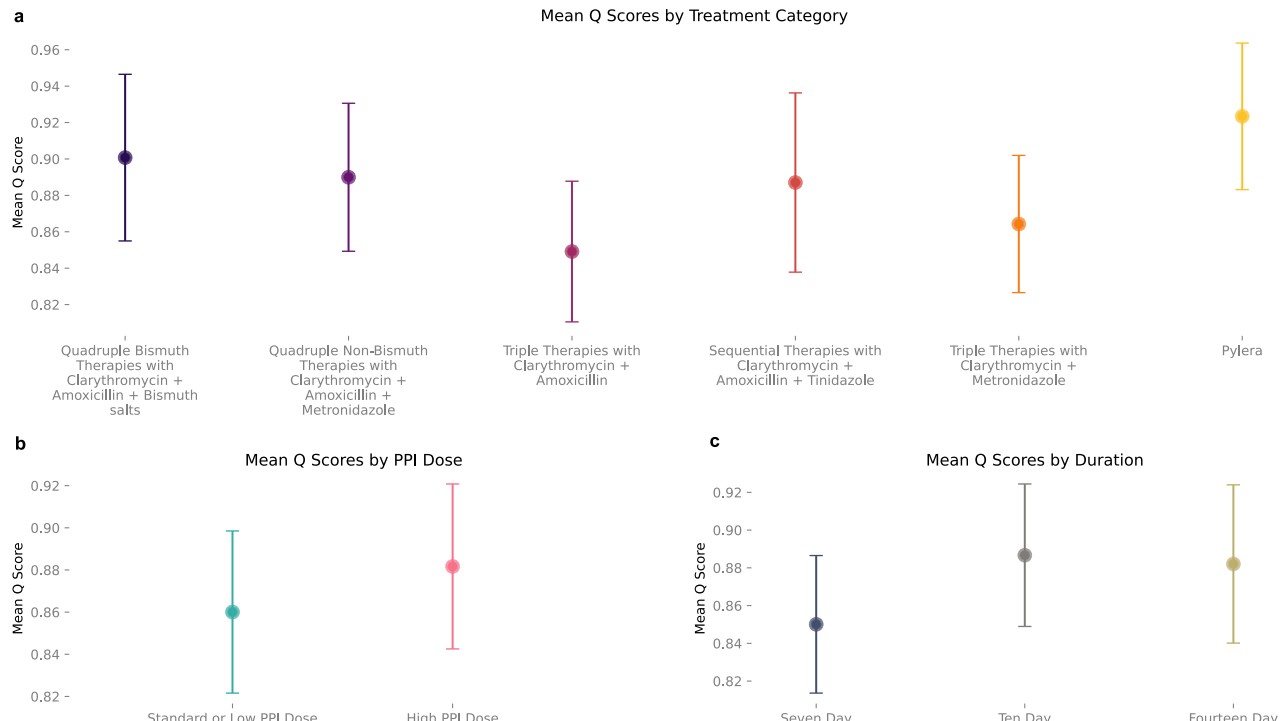

**Fig. 2 | *H. pylori* AI-clinician training and performance on real-world data. a** The mean Q scores of each treatment category in the testing phase (*n* = 3805 samples total) are compared, demonstrating the AI Clinician's preference for treatments on average. Pylera (yellow) has the highest overall Q score (mean = 0.92, SD = 0.04), followed by quadruple bismuth therapies (dark purple; mean=0.90, SD = 0.05), quadruple non-bismuth therapies (medium purple; mean = 0.89, SD = 0.04), and sequential therapies (red; mean = 0.89, SD = 0.05). triple therapies have the lowest Q score on average (mean = 0.86,0.85; SD = 0.04,0.05 for clarithromycin +

metronidazole, shown in orange, and clarithromycin + amoxicillin Therapies, shown in dark pink, respectively). All treatments include the prescription of a PPI. **b** Mean Q Scores by PPI Dose demonstrate High PPI dose (pink; mean = 0.88, SD = 0.04) has a higher average Q score than standard or low dose PPI (blue; mean = 0.86, SD = 0.04). **c** Mean Q scores by duration of treatment also demonstrate that 10 and 14 (khaki) day durations (light grey; mean = 0.89 and 0.88; khaki; SD = 0.04 and 0.04, respectively) out-perform 7 day (dark grey; mean = 0.85, SD = 0.04), though result in similar Q scores compared to one another.

first-line treatments from the Hp-EuReg dataset, with testing performed on the remaining 10%. AI performance was compared to clinicians by comparing Q scores of the clinician's action to the AI-recommended action for each patient. In a representative model, mean Q scores in the testing phase were tabulated to quantify the AI Clinician's preference for different treatment categories. All therapies include prescription of a PPI. Overall, Pylera® therapies had the highest average Q score (mean = 0.92, SD = 0.04), suggesting that over the entire testing dataset it was on average estimated to be the most effective on a diverse population of patients. It was followed by quadruple bismuth therapies with clarithromycin, amoxicillin, and bismuth salts (mean = 0.90, SD = 0.05), quadruple non-bismuth therapies with clarithromycin, amoxicillin, and metronidazole (mean = 0.89, SD = 0.04), and sequential therapies with clarithromycin, amoxicillin, and tinidazole (mean = 0.89, SD = 0.05). Triple therapies performed the most poorly, with clarithromycin and metronidazole (mean = 0.86, SD = 0.04) slightly outperforming clarithromycin and amoxicillin (mean = 0.85, SD = 0.05). (Fig. 2a) Average Q scores based on PPI dose demonstrate a preference for high dose PPI on average (mean = 0.88, SD = 0.04) compared to low or standard doses (mean=0.86, SD = 0.04). (Fig. 2b) For a full description of PPI dose definitions given particular PPIs, see "Methods". It is worth noting that many patients in the training dataset had a PPI dose which was not specified by the clinician. However, the category of patients with unspecified dose also showed high Q scores (mean = 0.89, SD = 0.04) suggesting this population was dominated by high PPI doses. Finally, ten (mean = 0.89, SD = 0.04) and fourteen-day (mean = 0.88, SD = 0.04) durations were found to have higher Q scores than 7-day (mean = 0.85, SD = 0.04) treatment periods—with ten- and fourteen-day periods showing similar average Q scores. (Fig. 2c).

When tabulated over all repeats (50 recommendations per patient in the testing phase), 65.5% of patients were consistently recommended a bismuth therapy consisting of either Pylera® or clarithromycin, amoxicillin, and bismuth salts paired with a PPI by more than half of the models, which we treat as a frequency threshold. Further, 15.5% of patients were recommended a non-bismuth clarithromycin, amoxicillin, and metronidazole with PPI, and 19.0% of patients recommended a variety of therapies, with no single therapy being recommended by more than half of models. Notably, no patients were consistently recommended triple or sequential therapies. (Fig. 3a) We additionally examined the breakdown of patients where unique bismuth therapies are considered separately, where patients need to be recommended either Pylera® or quadruple bismuth therapies with clarithromycin, amoxicillin, and bismuth salts by more than half of models to be considered above frequency threshold. 51.5% were recommended quadruple non-bismuth therapy with clarithromycin, amoxicillin, and metronidazole, 30.4% were recommended Pylera®, and 18.1% were recommended quadruple bismuth therapies with clarithromycin, amoxicillin, and bismuth salts. (Fig. 3b) As it is impossible to know the success rate of two different therapies on the same patient, the effectiveness of the AI Clinician was measured in a retrospective manner using the real-world treatments prescribed by clinicians. To evaluate the predicted performance of the AI Clinician in practice, we calculated the success rate of all treatments where the AI recommendation agreed with the treatment the clinician prescribed versus those where it did not and performed bootstrapping to construct a 95% confidence interval. Overall, treatments recommended by the AI Clinician showed a 94.1% success rate (*n* = 2988; CI: 93.2%, 95.0%) compared to treatments not recommended by the AI Clinician which showed a 88.1% success rate (*n* = 35,061; CI: 87.8%, 88.4%),

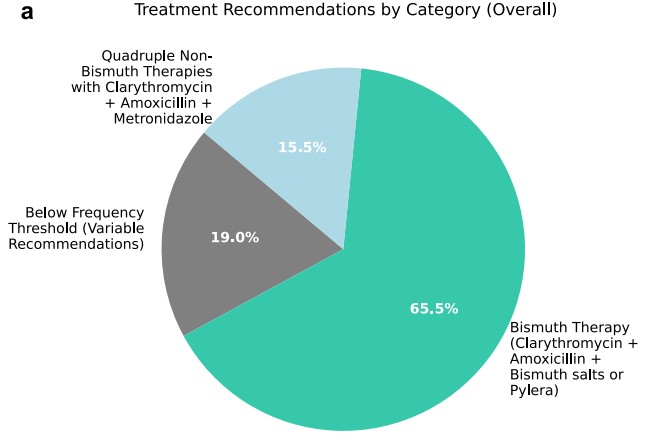

**a** Treatment Recommendations by Category (Overall)

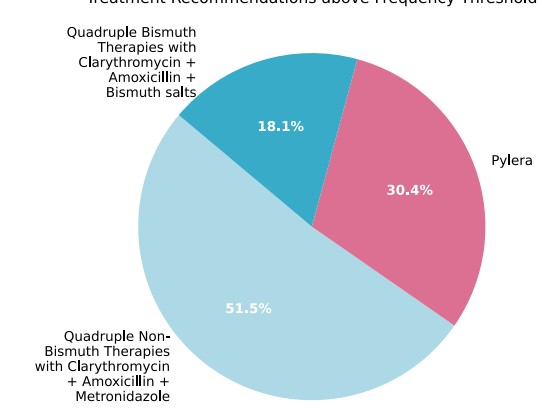

**b** Treatment Recommendations above Frequency Threshold

**c**

| | Pylera | Bismuth Quadruple | Bismuth Salts (Any) | Non-Bismuth Quadruple |
|---|---|---|---|---|
| Highly Associated Variables | Southwest Region | Taking Concurrent Medication (Any) | Eastern Region | Eastern Region |
| | Not taking Acetylsalicylic Acid | Eastern Region | Taking Concurrent Medication (Any) | Caucasian |
| | Taking Concurrent Medication (Any) | Not taking probiotics | Not Experiencing Heartburn | Not taking Rebamipid |
| Balanced Accuracy (%) | 84.7 | 76.7 | 73.6 | 92.3 |

**Fig. 3 | Personalized recommendations. a** Recommendations per individual across 50 repeated training-testing cycles on various splits of the data were generated, with mode treatment category tabulated for each patient, with the requirement that it was recommended by more than half of the repeats. On average, 65.5% of patients were recommended a Bismuth Therapy consisting of either Pylera® or clarithromycin, amoxicillin, and bismuth salts (with PPI) (represented in teal), 15.5% of patients were recommended non-bismuth quadruple therapy with clarithromycin, amoxicillin, and metronidazole (with PPI) (light blue), and 19.0% were recommended variable treatments, with no majority recommendation (grey). **b** When Pylera® and quadruple therapy with bismuth salts are distinguished, 30.4% of patients are recommended Pylera® more than half of the time (pink) and 18.1% are recommended quadruple therapy with bismuth salts (dark blue), suggesting that there is not a strong preference for which of the two therapies was recommended for the majority of patients which are routinely recommended a bismuth therapy. **c** Random Forest models are generated for each of the therapy categories discussed above to discover the relevance of patient variables in determining recommended therapy. Balanced accuracy measures the accuracy of the RF model in predicting which treatment a patient will be recommended. Variables are ranked by mean decrease in impurity (MDI) to determine the top three variables most associated with a particular treatment recommendation.

resulting in a net gain of 6.0%, demonstrating an association of AI-recommended therapies to eradication success. We further validated the performance of our model using an external dataset of 7186 patients, demonstrating a success rate of 92.8% ($n = 128$; CI: 88.4–97.1%) for treatments recommended by the AI Clinician compared to 87.4% ($n = 7048$; CI: 86.7–88.2%) for those that were not. (for full description, see *Supplementary Notes–Model Validation*.).

Finally, in order to see which variables correlated most strongly to treatment recommendation, we perform RF analysis each time for four recommendation groups: Pylera®, bismuth quadruple (with clarithromycin, amoxicillin, and bismuth salts), bismuth salts (any) which could be either Pylera® or bismuth quadruple therapy, and non-bismuth quadruple therapy (with clarithromycin, amoxicillin, and metronidazole). The model is formulated to predict whether a patient will receive a given treatment versus all others, based on patient variables. RF models had a balanced accuracy of prediction of 84.7% for Pylera®, 76.7% for bismuth quadruple therapies, 73.6% for bismuth salts (any), and 92.3% for non-bismuth quadruple therapies. Patient variable importance was ranked by mean decrease in impurity (MDI) to determine highest predictive power for a given treatment recommendation. Overall, being from the southwest region of Europe, not taking acetylsalicylic acid, and taking concurrent medication of any kind were more likely to result in a Pylera® recommendation. Taking concurrent medication, being from an eastern region of Europe, and not taking probiotics were more likely result in a bismuth quadruple recommendation. Likewise, being from an eastern region, taking any concurrent medication, and not experiencing heartburn as a symptom were more likely to correspond to a bismuth salts therapy of any sort. Finally, being from the eastern region, Caucasian, and not taking rebamipid were more likely to correspond to a recommendation of non-bismuth quadruple therapy. (Fig. 3c).

## Discussion

The *H. pylori* AI-clinician was developed to determine if AI-driven personalized treatments would boost treatment success compared to clinician-prescribed treatments alone, therefore benefitting patients. We found that this was the case, boosting success rate of prescribed therapies by 6.0% up to 94.1% for therapies recommended by AI from 88.1% for therapies that were not. Overall, we found that Q scoring in individual models was in line with current trends in treatment recommendations[14,39], demonstrating the reliability of the AI Clinician method. Pylera® and quadruple therapies with clarithromycin, amoxicillin, and bismuth salts or metronidazole, and sequential therapies showed the highest quality estimate by the AI clinician, in that order (and above triple therapies containing clarithromycin and amoxicillin or metronidazole). We also found that higher dose PPIs performed better than low and standard dose on average, suggesting most patients would benefit from a higher dose. Finally, while 10- and 14-day

durations out-performed 7-day, they performed quite similarly in terms of quality estimate to one another, likely driven by Pylera®'s increased effectiveness and 10-day formulation.

We found that 65.5% of patients were recommended a bismuth therapy by the majority of AI Clinician models trained on differing splits of data, while 15.5% were consistently recommended a non-bismuth quadruple therapy with clarithromycin, amoxicillin, and metronidazole. Overall, RF modelling was able to achieve a high balanced accuracy for the latter therapy, indicating that variables including presence in an eastern region, being Caucasian, and not taking rebamipid were highly indicative of a patient receiving a non-bismuth quadruple therapy recommendation. Pylera® was more likely to be recommended if a patient was from a southwest region and taking concurrent medication, but not acetylsalicylic acid. Non-bismuth quadruple therapies including clarithromycin, amoxicillin, and metronidazole were more likely to be recommended again if taking a concurrent medication, but instead from an eastern region and not taking probiotics.

The correspondence to region in personalized recommendations suggests several possibilities: that strains of *H. pylori* varying by region are driving the trend, that interactions between genetics and treatment are responsible, that trends in lifestyle varying by region are responsible, or a combination of the three. Further investigations to study a higher number of region-specific variables in detail will be needed to determine the explanation. Interestingly, though Pylera® did see the highest average Q score from modelling, it was not the most frequently recommended therapy overall (which was non-bismuth quadruple therapies). This likely suggests that these patients would receive a greater marginal benefit from recommended Pylera® treatment than other groups.

Interestingly, 19.0% of patients were recommended a variety of treatments, with no single category being recommended by more than half of models. In addition, most patients were not recommended the same treatment by every model generated by differing splits of training data. A further shortcoming of this study is that several treatment formulations (for example, the quadruple bismuth therapy including metronidazole, tetracycline, and bismuth subcitrate with PPI-which the single capsule Pylera® is based on) were not present in sufficient numbers in the dataset to be included in training due to the requirement of around 500 samples for stabilization in the network observed in our study. PPI dosage category was also divided into only two categories ('Standard or Low and High') to reduce the number of therapy type subdivisions due to the limited number of samples of each treatment category, though there are well-documented differences in effect between low and standard PPI doses. These facts signal the need to collect additional data for model training and for the improvement of the sensitivity of the AI Clinician in the future. Finally, all internal and external validation results documented in this work result from use of retrospective data, and therefore the conclusions of the study should be further validated by a prospective study in the future. We emphasize that this study does not necessarily demonstrate the superiority of AI over clinical decision-making, rather an improvement in recommendations that could be made by AI-assisted clinical decision-making.

Though these results are encouraging in terms of increasing early eradication of *H. pylori*, future work should be focused on treatment of patients for which the damage of long-term infection has already been done. For example, a well-established point of no return for infections exists, past which gastric cancer often develops even after eradication of *H. pylori*. Investigating other data types such as endoscopy images and omics data, especially with advanced methods such as ML and AI will be crucial for determining what specific consequences of infection a patient is experiencing or at risk of experiencing, and what therapeutic strategies may be applied. The *H. pylori* AI Clinician is intended for use alongside clinicians, and future methods would benefit from an approach which integrates rather than replaces human expertise.

With over half the globe experiencing *H. pylori* infection at some point in their lifetime, there is a great need to apply advances in AI to evaluate and improve management and treatment, especially with regard to treatment recommendation standards. This work demonstrates the robustness of current recommendation standards throughout a diverse and heterogeneous population, supporting their broad administration. Further, this work demonstrates a fundamentally novel system for making personalized treatment recommendations based on patient data, opening the door to many potential future applications.

## Methods

### Ethics statement
The Hp-EuReg study was conducted in accordance with the ethical principles outlined in the Declaration of Helsinki (1975, and its subsequent revisions) and complied with all relevant institutional and national ethical regulations. The study protocol was reviewed and approved in 2012 by the Ethics Committee of the Hospital Universitario de La Princesa (Madrid, Spain), which served as the reference Institutional Review Board (IRB). The protocol was also classified by the Spanish Agency for Medicines and Health Products (AEMPS), and prospectively registered at ClinicalTrials.gov under the identifier NCT02328131 (https://clinicaltrials.gov/study/NCT02328131).

### Method development
The *H. pylori* AI-clinician was developed in order to predict optimal treatment outcomes on a patient-specific basis while learning from real-world clinical decisions. It was implemented using RL, which is a machine learning approach where the so-called agent learns to maximize the reward it receives from taking actions in a trial-and-error manner. The agent was trained using clinical actions (treatments) and their observed reward (success/failure) from real-world patients previously recorded in Hp-EuReg database. (Fig. 4a) It should be stressed though that at this stage the AI-clinician is not being used nor should be used directly for the real-world clinical decision making for the new patients as it is still under development and is not certified as a medical device. RL was chosen over alternative formulations including logistic regression (LR), RF, and SVM-based models due to its observed superior performance as described in *Supplementary Notes−Comparison to Other Models*. Our approach is a treatment policy optimization task rather than one aimed at predicting labels. Since traditional formulations of LR, RF, and SVM are appropriate for tasks such as label prediction, and RL methods such as deep-quality network learning (DQN) have been designed to predict the quality of a multitude of possibilities during a decision-making task RL was an intuitive choice for further formulation. All software was developed in Python, relying mainly on scikit-learn[40] and PyTorch[41] packages for ML and AI modelling.

The specific RL method developed was termed independent-state Deep Q-Network Learning. (isDQN) It is an adaptation of Deep Q-Network (DQN) Learning, which is one of the most widely applied methods in RL[42–45]. Q-learning is a method by which the quality of state-action pairs is learned over time, where the state is a set of environmental variables and the actions are the set of all possible actions an agent may take. The action with the highest Q-score is chosen to be taken given the state context. DQN is termed deep in that it utilizes a neural network of several layers to achieve its learning. Reward is observed for each action in the context of a given state, and this reward is in turn used to alter weights in the neural network during some optimization step to improve the estimation of Q-scores in the future, therefore improving decision-making. Optimizations are some functions of both immediate reward from action taken, and the quality of the state the action puts the environment in the subsequent moment

**a** *H. Pylori* AI-Clinician Overview

**b** Hp-EuReg Data Preprocessing

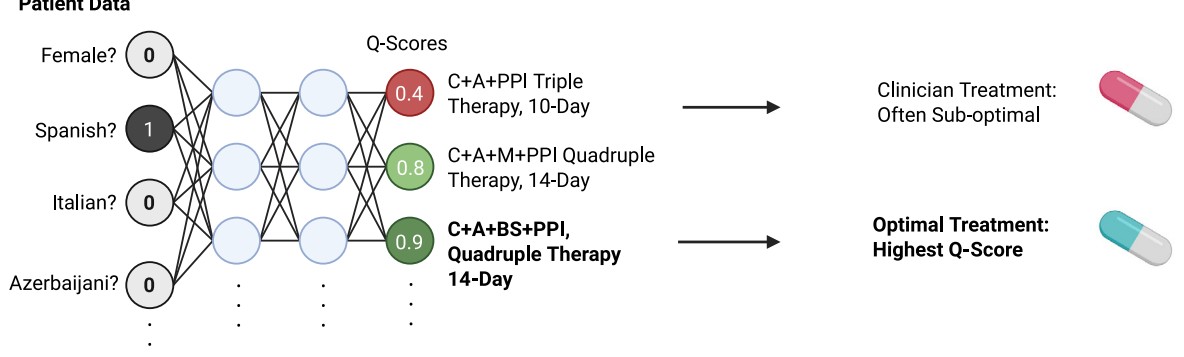

**c** Deep Quality Network (DQN) Overview

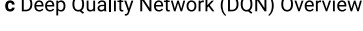

**Fig. 4 | *H. pylori* AI clinician methods overview. a** The *H. pylori* AI-Clinician is a recommendation system trained via reinforcement learning (RL) which learns from the result of clinical decision-making. Mathematically, the AI learns the quality of state, action pairs by observing the reward obtained by each. In practice, states are represented by patient data, and actions are represented by clinical treatment decisions. Reward is measured by the success or failure of treatment. Once trained, the AI-Clinician returns an optimal treatment for individual patients. **b** Hp-EuReg data, including 73,313 patient records and 30 pre-treatment patient variable categories, is preprocessed prior to model training. Only first-line treatments and patients who complied with treatment are considered resulting in 52,801 samples. Patient/Treatment pairs with less than 500 samples for a given treatment are removed to ensure sufficient training data. Actions (treatments) are encoded to include treatment category, antibiotics included, doses of antibiotics, and PPI Dose Category (including Low, Standard, High, and Other). State (patient) variables are one-hot encoded, resulting in 39,049 samples containing 77 patient variables total. **c** Deep Quality Network (DQN) analysis is implemented to train our recommendation system to identify optimal treatments. One-hot encoded patient data is fed into the network, which is followed by feeding into two hidden layers, and finally an output layer which represents the quality of implementing a given treatment for all possible treatments. The treatment with the highest quality (Q-Score) is the optimal treatment for a given patient. The network is trained via gradient descent at select optimization intervals (every 100 patients, optimized on previous 10 K patient states which are retained in model's memory for re-sampling at all times). During testing, patient state information is fed into the model to receive optimal treatment recommendations, but no further optimizations are performed. Created in BioRender. Higgins, K. (2025) https://BioRender.com/z5uh3k3.

(for example, increasing score in a video game immediately, and also putting the agent in a position to further increase score in future moves.).

Our method, isDQN differs from traditional RL methods in that there is no concept of a subsequent state for patient data as this data remains static before and after treatment, with eradication fully represented in the reward. Mathematically, optimization is performed using a loss function which aims to represent and minimize what cost was accrued by the current failures in decision-making at a given step. This loss function is calculated using the Q score of the current state subtracted from the immediate reward and maximum expected quality of the subsequent state. In our method, the quality of the subsequent state is not considered, effectively eliminating it from the loss function. (In mathematically rigorous terms, the trajectory is always taken to be in its final state, which is a special case described in the original formulation of DQN analysis detailed by Mnih et al[46].) For a demonstration of the model's validity and robustness using simulated data, see *Supplementary Material—Performance on Simulated Data*.

The patient information was used as follows. States were represented by the 77 one-hot encoded patient variables from the Hp-EuReg dataset. In brief, one-hot encoding is a method for representing categorical variables as binary vectors where membership of the sample in each of the possible categories is represented as a separate binary vector (true/false). This is needed to prevent the model from interpreting categorical data as ranked or continuous, which is essential in our case to avoid spurious relationships in modelling. Actions were

also one-hot encoded as described by clinical treatment decisions including antibiotic/PPI combination, antibiotic dose, PPI dose category (see *PPI Dose Mapping* for values), and duration (for example, clarythromycin + amoxicillin + metronidazole + clarithromycin dose = 500 mg, + intakes twice daily + … + high dose PPI, 14-day duration) and represented numerically. Only treatments with at least 500 examples in the dataset were included in the action space, resulting in 23 treatment options total. (Fig. 4b) Note that PPI dose category was divided into categories of 'Standard or Low Dose' ($n = 9935$) and 'High Dose' ($n = 16,304$−other doses not recorded) to reduce the number of treatment category divisions. Reward was represented by the clinical outcome of the treatment, therefore a value of +1 if eradication was achieved and −1 if the eradication was a failure (88.56% success rate overall). The model was trained using patient data in batches. Gradient descent was used to recalculate weights of the neural network at each optimization step using a mean squared error (MSE) loss function. Over time, this agent learns the quality of each treatment for a given patient, determining patient-specific optimal treatments. (Fig. 4c).

## Coding environment and computational resources
All code was written in Python 3.8.19, with Jupyter core version 5.1.0. pandas 2.0.3, Numpy 1.24.1, scikit-learn 1.3.2, and PyTorch 2.4.1+cu124 were also used for variable preprocessing, statistical, ML, and AI modelling. Analysis was performed on Intel64 Family 6 Model 165 CPU (Windows 10 OS) with 12 cores and total RAM of 15.79. No GPU resources were necessary for modelling.

## Variable preprocessing
The Hp-EuReg Dataset was obtained on February 14th, 2024 and was taken from the AEG-REDCap platform. Data were recorded in an Electronic Case Report Form (e-CRF) using the collaborative research platform REDCap hosted at Asociación Española de Gastroenterología, a non-profit Scientific and Medical Society focused on Gastro-enterology research[47]. The dataset consisted of 73,313 patients and 321 patient variables, including treatment administered and outcome in terms of *H. pylori* eradication. Samples were filtered to include only first-line treatments administered to patients who complied with their full regimen ($n_{samples} = 52,801$). In order to ensure sufficient training data for each treatment, treatments with <500 samples were also removed from the dataset (and therefore samples to which these treatments were administered, resulting in $n_{samples} = 38,049$). Missing values for any variable were encoded as NA and no samples with missing values for either eradication outcome or treatment were found in the remaining dataset. Variable preprocessing was performed to achieve a format suitable for one-hot encoding, with numeric variables also treated as categorical. Age was then binned into four categories: Under 40 ($n = 9779$), 40–50 ($n = 8008$), 50–60 ($n = 8560$), and Above 60 ($n = 11,682$). Countries were also categorized into regions, with East-centre ($n = 8588$), East ($n = 5975$), West-centre ($n = 2623$), North ($n = 1542$), South-west ($n = 18,662$), and Other ($n = 659$). (For a full list of countries and corresponding regions, see Supplementary Table 1.) Finally, variables were filtered to remove those which were reflective of treatment outcome or irrelevant to patient outcomes. Variables were one-hot encoded for upstream analysis. (Including dose mapping variables as described below, patient variable total $n_{samples} = 38,049$, $n_{one-hot\_encodings} = 77$).

## PPI dose mapping
PPI doses were mapped to provide a structured framework for interpreting dose variability between different PPIs. Omeprazole doses of 80 mg are considered High Dose, whereas doses of 10, 20, and 40 mg are considered Standard or Low Dose. For Lansoprazole, doses of 60 mg are considered high and 30 or 15 mg are considered standard or low. For Pantoprazole, all doses are considered standard of low (including 20 and 40 mg), whereas for Esomeprazole, 40 and 80 mg

are considered high, with 20 mg considered standard or low. Finally, Rabeprazole doses of 40 mg are considered high, whereas 20 and 10 mg are considered standard or low. Only 501 patients were given a dose deviating from these values, which we consider non-traditional and perhaps mis-entered, omitting this information and grouping with unlisted PPI values. Overall, 16,304 patients were given PPI doses considered High Dose, 9935 were given doses considered Standard or Low Dose. A large fraction of the dataset did not have a specified PPI dose by clinician, and when combined with non-traditional doses 11,810 termed Nondescript Dose were included in the dataset.

## Action space definition
The action space was defined in terms of antibiotic/PPI combination prescribed by the clinician as well as the dose of each medication, number of intakes (for example, twice a day), PPI dose category, and duration it was prescribed. For example, a quadruple bismuth regimen of clarithromycin (dose=500 mg, twice daily), amoxicillin (1000 mg, twice daily), bismuth salts (120 mg, four times daily), and high dose PPI for a 14-day duration would be represented as a single numeric action. The action space was restricted to include only treatments with at least 500 examples in the dataset.

## Independent-state deep quality network learning (isDQN)
The analysis method used to train our recommendation system, termed independent-state Deep Quality Network Learning (isDQN) takes the form of a traditional DQN analysis, except that optimizations to network weights do not consider a subsequent state when calculating loss. A deep neural network of four layers was implemented for this analysis, consisting of an input layer ($n_{nodes} = 77$, the number of one-hot encoded variables for each patient), two hidden layers ($n_{nodes} = 128$ each), and an output layer ($n_{nodes} = 23$, the total number of combinations of treatment and duration). Inputs to the optimization step consist of:

- $\boldsymbol{\phi}$, the state space, where $\boldsymbol{\phi} = \{\boldsymbol{\phi}_1, \ldots, \boldsymbol{\phi}_{n_{state}}\}$ and represents the set of binary (one-hot) encoded patient variables, and $n_{state} = 77$.
- **A**, the action space, where $\mathbf{A} = \{a_1, \ldots, a_{n_{action}}\}$ and represents the numerically encoded combination of treatment and durations of antibiotic combinations with durations of 7, 10, and 14 days, and therefore $n_{action} = 23$.
- **R**, the reward space, where $\mathbf{R} = \{-1, +1\}$ and +1 represents successful eradication, whereas −1 represents failed eradication.

Optimization was performed via gradient descent on a smooth L1 loss function, which was chosen due to its lessened sensitivity to outliers and tendency to avoid exploding gradients. For a batch size $N$, loss is defined as:

$$l(x,y) = \text{mean}(\mathbf{L}) = \text{mean}(\{l_1, \ldots, l_N\}^T) \tag{1}$$

Where:

$$l_n = \frac{(x_n - y_n)^2}{2 * \beta} \text{ if } |x_n - y_n| < \beta, \ |x_n - y_n| - \frac{\beta}{2} \text{ otherwise} \tag{2}$$

Where $\beta = 1$, and $N$ = number of patients per batch, and

$$x_n = R_n \tag{3}$$

Where $R_n$ is the reward observed for patient n,

$$y_n = Q(\boldsymbol{\phi}_n, a_n; \boldsymbol{\theta}) \tag{4}$$

Where $Q(\mathbf{s}_n, a_n)$ is the quality score for the state-action (patient-treatment) pair of patient $n$ determined by a forward pass through the neural network and $\boldsymbol{\theta}$ are the neural network parameters.

The isDQN algorithm is modified from the original DQN algorithm detailed in Mnih et al.[46] in **Algorithm 1**.

## Selection of optimal treatment for AI recommendation

The AI-recommended action for a given patient is defined by:

$$a^*(\boldsymbol{\phi}_n) \leftarrow \mathrm{argmax}_a Q(\boldsymbol{\phi}_n; \boldsymbol{\theta})$$

**Algorithm 1.** : Independent-State Deep Quality Network Learning (isDQN)

> Initialize replay memory D to capacity N
> Initialize action-value function Q with random weights $\boldsymbol{\theta}$
> **For** episode = 1, M **do**
>> Initialize sequence $s_1 = \{x_1\}$ and preprocessed sequence $\boldsymbol{\phi}_1 = \boldsymbol{\phi}(s_1)$
>> **For** t = 1,T **do**
>>> Store transition ($\boldsymbol{\phi}_t$, $a_t$, $r_t$) in D
>>> Every b steps **do**
>>>> Sample random minibatch of transitions ($\boldsymbol{\phi}_j$, $a_j$, $r_j$) from D
>>>> Set $y_j = r_j$
>>>> Perform a gradient descent step on $\ell(y_j, Q(\boldsymbol{\phi}_j, a_j; \boldsymbol{\theta}))$ with respect to the network parameters $\boldsymbol{\theta}$
>> **End for**

Where the argmax function determines the state-action pair of highest Q value by considering the quality of all possible actions for a given patient, defined by patient variables $\boldsymbol{\phi}_n$. Therefore, Q-scores of AI actions can be directly compared the Q scores of Clinical decisions via $Q(\boldsymbol{\phi}_n, a^*(\boldsymbol{\phi}_n); \boldsymbol{\theta}))$ and $Q(\boldsymbol{\phi}_j, a_j; \boldsymbol{\theta})$, respectively.

Where $\boldsymbol{\theta}$ are the network parameters, $\boldsymbol{\phi}$ represents the state, $a$ represents the action, and $r$ represents the reward. D represents a batch of memories D with capacity N. Q represents the action-value function, M represents the number of episodes, and T represents the number of transitions. $\ell$ represents the loss function for which we perform a gradient descent step with, taking the true outcome and quality estimate of state-action pairs as input.

## Single model examination

To examine average trends, Q scores were broken down by treatment categories including quadruple bismuth therapies with clarithromycin, amoxicillin, and bismuth salts, quadruple non-bismuth therapies with clarithromycin, amoxicillin, and metronidazole, sequential therapies with clarithromycin, amoxicillin, and tinidazole, triple therapies with clarithromycin and amoxicillin, triple therapies with clarithromycin and metronidazole, and Pylera®. All treatments include a PPI in the formulation. Treatments are also broken down by PPI dose into categories of low/standard and high. The subset of patients recommended a nondescript PPI dose was dropped from this analysis. Finally, Q scores are examined on average by duration, including seven-, ten-, and fourteen-day formulations. Mean and standard deviation for each of these groups was calculated to achieve a comparative ranking of each treatment component.

## Patient specific recommendation analysis

Ten-fold, fifty repeat cross-validation was chosen to evaluate the quality and consistency of our analysis. In ten-fold cross validation, the dataset is divided into ten random samples or folds. Iteratively each fold is taken to represent a testing subset of the data, whereas the remaining nine are taken as a training subset, therefore applying a 90−10 percent training-testing split. Network training (optimization) is performed via isDQN analysis as described above, where the quality of AI and clinical decision is assessed via Q-score at each patient iteration. During testing, no further optimizations are performed and only the quality of AI and clinical decisions are assessed. This process is repeated fifty times to check for bias in random samples and to evaluate the most common recommendation for each patient after many repeats. The recommended treatments in the testing phase of each model were tabulated for each patient, resulting in 50 recommendations total per patient. The mode treatment for each person was recorded to examine heterogeneity in the treatment recommendation. To evaluate the predicted performance of AI recommendations, the success rate of all treatments which agreed with those prescribed by real world clinicians was compared to those which disagreed. A 95% confidence interval was conducted by bootstrapping 1000 times.

## Determining relationship of patient variables to treatment recommendation by random forest

RF models are generated for each of four treatment categories: Pylera®, bismuth quadruple (including clarithromycin, amoxicillin, and bismuth salts), bismuth salts (any)−which includes either Pylera® or bismuth quadruple therapy in the formulation described above, and non-bismuth quadruple therapy including clarithromycin amoxicillin, and metronidazole. The RF model is formulated to predict whether a patient will receive a given treatment versus all others. Patient variables are then ranked to determine size of effect in the model on predicting outcome using a metric of mean decrease of impurity (MDI). Full lists of ranked variables and MDI are available in Supplementary Data 2.

## Theoretical AI clinician testing

In addition to real-world clinical data, synthetic dataset of 10,000 patients with 100 binary variable features was generated for model stress testing and verification. To model a dataset with imbalanced classes, patients were split into two groups of 7000 and 3000 patients, respectively, where binary variables were identical within groups by a random selection of 0's and 1's. Next, noise was added to the dataset by introducing a random selection of a 0 or 1 at random intervals in each patient to a desired level. Datasets with noise levels of 5, 10, 25, 50, 75, and 99% were generated. Each group of patients was 'treated' half of the time with treatment A and half of the time with treatment B, where treatment A was made 90% effective in group A and 80% effective in group B. Treatment B was made 90% effective in group B and 80% effective in group A. Rewards of +1 were assigned for successful treatments and −1 for failures at the 90% and 80% rate described above. Training was performed on a balanced randomized 90-10 training-testing split, where optimization was carried out using Mean Square Error (MSE) loss and hyperparameters of batch size 1000, learning rate 0.001, deque size of 1000, and steps to optimize of 100. Evaluation of results was performed by counting the number of times a treatment with 90% effectiveness in its respective group was recommended to a patient from this respective group, for example, treatment A recommended to a patient from group A.

## Sensitivity analysis and hyperparameter selection

To determine optimal hyperparameters in the context of a more complex dataset similar to Hp-EuReg, 50,000 patients were generated with 10 unique types of 70 patient variables. For each of the ten groups, one treatment was given 90% effectiveness while all others were given 70% effectiveness, with no treatment being 90% effective in any two groups of patients. A hyperparameter grid was tested including: batch size of 1000, 5000, and 10,000; a learning rate of $5*10^{-5}$, $10^{-4}$, $5*10^{-4}$, and $10^{-3}$; steps to optimize of 50 and 100; and a deque size of 1000, 5000, and 10,000. Scoring of best hyperparameters was based on the fraction of treatments correctly recommended to patient group based on higher efficiency in training data. The hyperparameters chosen for further modelling had the highest percentage of correct recommendations of 99.9%, and consisted of a batch size of 1000, learning rate of $5*10^{-5}$, and deque size of 1000.

## Recruitment Information

Overall, a total of 70,915 adult participants (40% males, 60% females; age range: 18–99 years; mean age: 50 years) were enroled in the study between (2013-March 2025). Sex was recorded based on participant self-report, and no gender identity information was collected. All participants provided written informed consent prior to enrolment, in accordance with institutional and national ethical standards. Participants did not receive financial compensation for their participation. Sex was considered in the study design primarily for descriptive and stratified analyses. However, no specific sex-based subgroup analyses were conducted as the study was not powered to detect sex differences, and the primary objectives did not include sex-disaggregated outcomes. Data have been disaggregated by sex where available and appropriate, and individual-level sex-disaggregated data are included in the Source Data file. All procedures and reporting comply with the SAGER (Sex and Gender Equity in Research) guidelines.

## Reporting summary

Further information on research design is available in the Nature Portfolio Reporting Summary linked to this article.

## Data availability

Source data are provided with this paper. The de-identified clinical data generated and analyzed in this study have not been deposited in a public repository due to European Union General Data Protection Regulation (GDPR) constraints and ethical considerations involving participant confidentiality. Processed and aggregated data underlying the findings are available under restricted access for reasons related to patient privacy and institutional data governance. Access can be obtained by submitting a research proposal to the Hp-EuReg Data Access Committee at opn.aegredcap@aegastro.es. Qualified academic or clinical researchers may be granted access for non-commercial, ethically approved purposes. Requests are reviewed within 21 calendar days, and data—if approved—will be shared via a secure platform for a period of up to 12 months. Raw individual-level data are protected and cannot be shared publicly in compliance with data privacy laws. Previously published datasets used in this study from the Hp-EuReg registry are available at: Hp-EuReg Publications. No datasets are available in the Supplementary Information.

## Code availability

Code has been made available at the following public Bitbucket repository: https://bitbucket.org/iAnalytica/aiclinician/src/main/.

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

## Acknowledgements

This research was supported by the AIDA project, funded by UK Research and Innovation (Grant No. 10058099) and the European Union (Grant No. 101095359). All authors and consortium members received funding through this project. O.N. and J.G. were additionally supported in the compilation of the Hp-EuReg database by the European Helicobacter and Microbiota Study Group (EHMSG), the Spanish Association of Gastroenterology (AEG), the Centro de Investigación Biomédica en Red de Enfermedades Hepáticas y Digestivas (CIBERehd), the European Union's Horizon Europe programme (Grant Agreement No. 101095359), UK Research and Innovation (Grant Agreement No. 10058099), and the European Union's EU4Health programme (Grant Agreement No. 101101252). The views and opinions expressed are those of the author(s) and do not necessarily reflect those of the European Union or the Health and Digital Executive Agency (HaDEA). Neither the European Union nor the granting authorities bear responsibility for the content. O.N. and J.G. acknowledge Diasorin, Juvisé, and Biocodex for providing funding to the Hp-EuReg study. However, these companies had no access to clinical data and were not involved in any stage of the study, including its design, data collection, statistical analysis, or manuscript preparation. We acknowledge their financial support with gratitude. We thank the Spanish Association of Gastroenterology (AEG) for providing the e-CRF service free of charge. Figures 1, 4, and S1A made in BioRender (www.biorender.com).

## Author contributions

K.H. was responsible for methodological development, simulation design and experiment, evaluation of results and manuscript preparation. K.V., D.V., I.L., and T.F.K. worked on conceptualization, study design, methodology formulation/developments and data analysis. Members of the AIDA consortium contributed to the project design and conceptualization. O.N. managed data collection, curation, and advising on processing and interpreting results. K.V. supervised method development, simulations, and evaluations. J.S. and I.L. provided critical advice on bioinformatic and machine learning analysis, including evaluation and interpretation of results. J.G. was responsible for overseeing data collection and interpretation of results. T.F.K. and K.V. were responsible for securing funding and overall project management. All authors contributed to the writing and editing of the manuscript and approved the final version.

## Competing interests

Javier P. Gisbert has served as speaker, consultant, and advisory member for or has received research funding from Mayoly Spindler, Allergan, Diasorin, Richen, Biocodex and Juvisé. Olga P. Nyssen received research funding from Allergan, Mayoly Spindler, Richen, Biocodex and Juvisé. Drs Kirill Veselkov, Ivan Laponogov, and Dennis Veselkov are affiliated with Intelligify Ltd, an AI consultancy company, which was not involved in the research, analysis, or interpretation of the results presented in this study. Tania Fleitas Kanonnikoff discloses advisory roles honoraria from Amgen, AstraZeneca, Beigene, BMS and MSD. Institutional research funding from Gilead. Speaker honoraria from Amgen, Servier, BMS, MSD, Lilly, Roche, Bayer. The remaining authors declare no conflicts of interest. POLICY DISCLOSURE-USE OF CLINICAL DATA. This study involves the secondary analysis of de-identified clinical data obtained from the European Registry on Helicobacter pylori Management (Hp-EuReg). The data were originally collected by the Hp-EuReg consortium across multiple centres in Europe under appropriate ethical approvals and patient consent at the time of collection. No new data were collected for the purposes of this analysis, and the authors were not involved in direct recruitment or interaction with study participants. All analyses were conducted on anonymised data in accordance with applicable data protection and ethical guidelines.

## Additional information

## AIDA CONSORTIUM

Tania Fleitas Kanonnikoff[4,5,21] ✉, Ana Miralles Marco[4], Manuel Cabeza-Segura[4], Elena Jiménez Martí[4,7], Josefa Castillo[4,5,7], Kirill Veselkov[1,6,21] ✉, Mārcis Leja[8], Inese Poļaka[8], Fatima Carneiro[9], Ceu Figueiredo[9], Rui M. Ferreira[9], Rita Barros[9], Javier P. Gisbert[2,21] ✉, Olga P. Nyssen[2,20], Leticia Moreira[10], Miriam Cuatrecasas[11], Gloria Fernandez-Esparrach[10], Tamara Matysiak-Budnik[12], Jerome Martin[12], Laimas Jonaitis[13], Juozas Kupčinskas[13], Paulius Jonaitis[13], Mário Dinis-Ribeiro[14], Miguel Coimbra[15,16], Ana Carina Pereira[14], Filipa Fontes[14], Manon C. W. Spaander[17], Judith Honing[17], Stefano Sedola[18], Junior Andrea Pescino[18], Zorana Maravic[19] & Ana Martins[19]

[7]Biochemistry and Molecular Biology Department, Universitat de València, Valencia, Spain. [8]Institute of Clinical and Preventive Medicine, Faculty of Medicine and Lifesciences, University of Latvia, Riga, Latvia. [9]i3S—Instituto de Investigação e Inovação em Saúde, Universidade do Porto; Ipatimup—Institute of Molecular Pathology and Immunology of the University of Porto; Faculty of Medicine of the University of Porto; Department of Pathology, Unidade Local de Saúde São João, Porto, Portugal. [10]Department of Gastroenterology, Hospital Clínic de Barcelona; Institut d'Investigacions Biomèdiques August Pi i Sunyer (IDIBAPS); Centro de Investigación Biomédica en Red de Enfermedades Hepáticas y Digestivas (CIBEREHD); Facultat de Medicina i Ciències de la Salut, Universitat de Barcelona (UB), Barcelona, Spain. [11]Department of Pathology, Hospital Clínic de Barcelona; Institut d'Investigacions Biomèdiques August Pi i Sunyer (IDIBAPS); Centro de Investigación Biomédica en Red de Enfermedades Hepáticas y Digestivas (CIBEREHD); Facultat de Medicina i Ciències de la Salut, Universitat de Barcelona (UB), Barcelona, Spain. [12]Institut des Maladies de l'Appareil Digestif, Hépato-Gastroentérologie, Hôtel Dieu, Centre Hospitalier Universitaire, Nantes, France. [13]Department of Gastroenterology, Lithuanian University of Health Sciences, Kaunas, Lithuania. [14]RISE@CI-IPOP (Health Research Network), Portuguese Oncology Institute of Porto (IPO Porto), Porto, Portugal. [15]Instituto de Engenharia de Sistemas e Computadores, Tecnologia e Ciência, Rua Dr. Roberto Frias, 378, 4200-465 Porto, Portugal. [16]Faculdade de Ciências, Universidade do Porto, Rua do Campo Alegre, s/n, 4200-465 Porto, Portugal. [17]Department of Gastroenterology and Hepatology, Erasmus University Medical Center, Rotterdam, the Netherlands. [18]StratejAI, Avenue Louise 209, Brussels, Belgium. [19]Digestive Cancers Europe, Rue de la Loi 235/27, Brussels, Belgium.

