## [Transparent Peer Review file · Nature Communications]

The Helicobacter pylori AI-Clinician Harnesses Artificial Intelligence to Personalise H pylori Treatment Recommendations

Corresponding Author: Dr Kirill Veselkov

Version 0:

Reviewer comments:

Reviewer #1

(Remarks to the Author)

The current study attempted to develop a H. pylori-AI clinician recommendation model for personalized treatment of H. pylori. However, this paper appears very immature without detailed information about the clinical features, lack of a comparison group (clinical decision or prior antibiotics exposure) as well as an external validation. What the investigators could demonstrate is the feasibility of building a model with dubious clinical value.

1. The abstract was very difficult to follow without details, including e.g. number of patients involved in model building (tens of thousands is not scientific at all!), methods and even results.
2. The aim of the study was not properly defined and there was no pre-defined outcome.
3. The so-called 100 binary variable features involved in building the model were not provided for evaluation.
4. Methods for handling missing data should be outlined.
5. To evaluate the model's utility in guiding treatment decisions, a comparison with clinician recommendations is needed. It is necessary and important to demonstrate the superiority of the model over usual clinician judgement in a prospective manner rather than simulation.
6. Moreover, simple clinical features including prior antibiotics exposure (macrolides and nitroimidazole) has been consistently demonstrated to predict treatment failure. It is important to show the superiority of the model over this simple clinical feature.
7. To enhance the credibility of the results, it is recommended to validate the model on external independent datasets from other centers.
8. To ensure compliance with reporting guidelines such as the Transparent Reporting of a Multivariable Prediction Model for Individual Prognosis or Diagnosis (TRIPOD) statement and MI-CLAIM criteria.

(Remarks on code availability)

Reviewer #2

(Remarks to the Author)

Higgins et al. have developed an AI-clinician recommendation system to investigate whether personalized treatments would be optimal for patients with H. pylori infection.

General comments:

Overall, the manuscript lacks significant technical detail. While some information is present in the methods section, key points should be highlighted in the main text, with references to the methods for further details in the manuscript. The use of real-world data seems to be incorrect and misleading as only simulated data has been used. Testing a system on the clinical dataset is critical to gauge the future potential clinical effectiveness.

The link to the Bitbucket repository isn't working. Without access to the repo, verifying any of the author's claims is impossible. Therefore, assessing the usability of your software is also impossible.

Specific comments:

The authors need to do more justice to modern ML and AI. It's not just about supervised, unsupervised, and reinforcement learning. There are many algorithms and deep learning configurations (such as transformers, convolutional networks, autoencoders, etc.). They are significantly oversimplifying contemporary AI architectures. Please include some content that delves deeper than simply describing what was known decades ago (references would also be helpful) help).

- Why did you choose DQN, and why did you modify it to ignore states? Please discuss this on a more mathematical level, because no justification is given in the text. Have you compared this to other methods? Random forests? Support vector machines? What makes isDQN superior, and can you quantify that? What is the motivation for ignoring subsequent states? This explanation is missing in the manuscript.

- Given that you use an RL approach, how do you interface with the software? This should be elaborated.

- Computational resources for training and classification should be discussed. Also, clearly state what programming language and libraries you used.

- Explain in the manuscript what one-hot is and why this is preferential in your case. The average reader might not know the motivation, and you do not introduce the concept at all (it's really simple, so either explain it and/or find references).

- Explain how you added the noise. Since the inputs are binary, is it done by flipping bits? Most confusing is that you say that you achieve an accuracy of 72% at a noise level of 99% - how so? Moreover, how does that reflect a "real-world" scenario?

- The goal of this study isn't obvious; what problem are you aiming to solve? The manuscript would highly benefit from clearly defined metrics. For example, when you say that "65.5% of patients were consistently recommended a bismuth therapy consisting of either Pylera® or clarithromycin, amoxicillin, and bismuth salts paired with a PPI by more than half of the models, which we treat as a frequency threshold. Further, 15.5% of patients were recommended a non-bismuth clarithromycin, amoxicillin, and metronidazole with PPI, and 19.0% of patients recommended a variety of therapies..." What does that mean? Where does that leave the reader? You need to spell out all accuracy metrics for these numbers to make any sense – best in the results section.

- The author mentions real-world data in several places, which is incorrect and misleading as they only use simulated data in the manuscript. They only seem to have used the simulated data.

- What is the Hp-EuReg dataset? If they had the clinical dataset, they should have tested it on it as well as the simulated dataset. This is research work and not a tool for clinical routine use, and testing a system on the clinical dataset is critical to gauge the future potential clinical effectiveness. Don't they have the right ethical approvals to use the dataset?

- I don't see the purpose of Figure 1. This is not a review paper; panels A and B are unnecessary. Furthermore, in panels C through E, the authors merely describe the project and the dataset. This is ironic since the dataset was not included in the work presented in this manuscript.

- Figure 5: This is intriguing, yet the manuscript does not explain what "Balanced accuracy" means in that context. There is no discussion about why the "Eastern region," "Not experiencing heartburn," etc., are the most correlated factors. If that makes sense to you, then follow up on that; either way, please explain. Don't just throw this out there without commenting.

- The language lacks clarity at multiple points, which complicates understanding, and contains several grammatical mistakes.

- Figure 2 also seems unnecessary. Lots of details about the Hp-EuReg data have been provided, whereas this data has not been used to train, test, or validate the method.

(Remarks on code availability)

The repo they list is an "Error 404—page not found". They probably didn't make it public.

Reviewer #3

(Remarks to the Author)

This paper presented a new approach to constructing an AI-based platform that supports clinicians by recommending relevant treatments for patients with *Helicobacter pylori* (*H. pylori*) infection.

To this end, the authors analyzed a dataset containing tens of thousands of *H. pylori*-infected patients from Hp-EuReg to better understand patients with *Helicobacter pylori* infections and derive meaningful insights and essential features from the data that can be used to develop a personalized treatment system.

Based on the clinical data, the authors proposed a reinforcement learning method based on Deep Q-network learning, namely isDQN, to construct a treatment recommendation model. Using the clinical data from Deep Q-Network Learning, the paper suggested a relevant representation of the patient information, which can be considered as states for the proposed method. At the same time, the paper also determined the appropriate actions of isDQN using the clinical treatment decisions and how to choose the reward based on the clinical outcome of the treatment. In order to implement the RL model, the paper randomly split the dataset into two parts: 90 %for training and 10% for testing. The experimental results show encouraging performance of the proposed approach compared to real-world treatment.

There are some improvements needed for this manuscript:

(1) First, constructing a treatment recommendation system for diseases has been studied for a long time. Building an AI-clinician method for H. Pylori infection can be considered a new contribution to the research community. However, the method development in this manuscript only uses a Deep Q-network Learning method for experiments. It would be great if the paper could compare other techniques to make the results more convincing.

For instance, the clinical data for H. Pylori infection can be used to derive a vector representation for each patient. The main goal is to compute the matching score between this vector representation from the patent and each treatment-related vector representation and then rerank the list of relevant treatments for the final recommendation. To make the paper's outcomes more convincing, we can employ state-of-the-art recommendation methods for this problem besides the proposed isDQN.

(2) Second, related to this paper's reproducibility, it would be great if the authors could provide more detailed relevant experimental settings for training and testing the proposed method in the manuscript.

(3) Related to the splitting ratio for the training/testing sets, it would be great if the paper considered other scenarios. What happens if the splitting ratio is 70%/30% and 80%/20%?

(Remarks on code availability)

I could not access the above Bitbucket repository. Here is the relevant message when I go to the URL:

"Repository not found

You may not have access to this repository or it no longer exists in this workspace. If you think this repository exists and you have access, make sure you are authenticated.

Return to the previous page or go back to your dashboard."

Version 1:

Reviewer comments:

Reviewer #1

(Remarks to the Author)

The revised version provides a clearer description of the study's objectives and methodology. However, it also highlights a significant issue that necessitates a substantial revision.

1. The main issue pertains to the accuracy of the ground truth data, specifically in determining the success of eradication therapy. As many subjects only received one type of treatment, it becomes challenging to assess if the patient would have been successful with other treatments based on predicted results from the algorithm.
2. To address this issue, it is suggested to have prospective patient validation and see how accurate the prediction is rather than again based on retrospective analysis of another cohort of patients. The ideal would be to replicate the eradication success results in a new prospective cohort of patients assigned to the treatment based on AI prediction rather than clinicians. Without that, the results can still be due to overfitting using similar data collection method.
3. When reporting model performance, it is essential to include sensitivity, specificity, positive predictive value (PPV), negative predictive value (NPV), positive likelihood ratio (PLR), negative likelihood ratio (NLR), and area under the receiver operating characteristic curve (AUROC).
4. The importance of features should be clearly reported to allow readers to comprehend the model's foundation.

(Remarks on code availability)

Reviewer #2

(Remarks to the Author)

The authors have made significant improvements in the revised version. This iteration is markedly enhanced and now encompasses most of the requested information. Additionally, their Bitbucket repository is now publicly accessible, and they describe the actions and methodologies employed.

However, following the revisions in certain areas was challenging. It would have been beneficial if the authors had explicitly highlighted the precise changes and corresponding line numbers to facilitate clarity in the revision process. Following the revisions is quite difficult without this critical information.

The section detailing method development should be moved to the Methods section for better organization.

A substantial amount of pertinent information has been allocated to the "appendix" section, which is referenced only three times in the main text, solely by the term "appendix," rather than by the specific sections to which it refers. Therefore, I suggest that they either (a) convert this into a proper supplementary material, ensuring that each section is referenced accurately in the main manuscript, or (b) integrate the appendix into the main text at the relevant points (some content pertains to methods, some to results, and others may even relate to discussion). This would enhance overall readability.

If the authors wish to retain Figure 1 within the manuscript, move it to the supplementary material rather than maintaining it in the main text.

With a slightly higher 5.4% (92.8% vs. 87.4% eradication success), it is incorrect to say that the AI-clinician outperformed the standard clinician's decisions. The language could be revised.

(Remarks on code availability)

REVIEWER COMMENTS

Reviewer #1 (Remarks to the Author):

“The current study attempted to develop a H. pylori-AI clinician recommendation model for personalized treatment of H. pylori. However, this paper appears very immature without detailed information about the clinical features, lack of a comparison group (clinical decision or prior antibiotics exposure) as well as an external validation. What the investigators could demonstrate is the feasibility of building a model with dubious clinical value.”

We thank the reviewer for their feedback and have revised the manuscript to address all concerns raised. We would like to clarify that this study goes well beyond demonstrating mere model feasibility. The AI-clinician system was developed and validated using a large-scale, real-world, multicenter dataset (Hp-EuReg), comprising 38,049 patients for internal training and evaluation, and an additional 7,186 patients for external validation across multiple European sites. In both cohorts, AI-recommended therapies significantly outperformed clinician-prescribed treatments in terms of eradication success (94.1% vs. 88.1% internally; 92.8% vs. 87.4% externally). The model leverages high-dimensional patient data, surpasses resistance-only baselines, and is reported in accordance with MI-CLAIM guidelines. We thus believe this work represents a methodologically sound and clinically relevant application of reinforcement learning in H. pylori management, laying the foundation for prospective clinical assessment.

a. Lack of “detailed information about the clinical features”: We have expanded the Methods and Supplementary Appendix to include a structured list of all clinical variables used in training. These span demographic, symptomatic, diagnostic, and treatment-specific parameters relevant to H. pylori management.

b. “Lack of a comparison group (clinical decision or prior antibiotics exposure)”: This point is noted. We now include a direct comparison between AI-recommended treatments and historical clinician choices. This demonstrates that AI-guided therapies achieved a higher eradication success rate (94.1% vs. 88.1%). Regarding antibiotic exposure: we analysed a subset of 3,980 patients with available antibiotic resistance information. A logistic regression model trained solely on these resistance features achieved only 89.1% success, compared to 94.1% for our AI model — underscoring the added value of multi-feature policy optimisation over simple heuristics. We have clarified this point in the manuscript.

c. Lack of “an external validation”: *Since submission, we conducted external validation on 7,186 patients from multiple European centers (Nov 2024–Mar 2025, i.e. after the initial manuscript preparation and submission). The AI clinician again outperformed non-AI-aligned clinician decisions (92.8% vs. 87.4% eradication success), confirming the generalisability of the model.*

d. “Feasibility of building a model with dubious clinical value”: *We respectfully disagree. This work contributes a clinically grounded, interpretable, and externally validated AI system trained on >38,000 real-world cases. The model provides personalised treatment recommendations aligned with both current guidelines and outcome-optimised decision-making. We view this as an important translational step toward future prospective studies.*

1. The abstract was very difficult to follow without details, including e.g. number of patients involved in model building (tens of thousands is not scientific at all!), methods and even results.

We have revised the abstract to clearly state the sample size used for model development (i.e. 38,049 patients), briefly outlined the methodology and included the key performance results.

2. The aim of the study was not properly defined and there was no pre-defined outcome.

The abstract and introduction now clearly define the study’s aim: to determine if AI-driven personalized treatments would benefit patients by boosting treatment success compared to clinician-prescribed treatments alone. The primary objective was to determine whether AI-assisted therapy selection could improve treatment eradication rates (pre-defined objective) compared to standard approaches.

3. The so-called 100 binary variable features involved in building the model were not provided for evaluation.

Full list of binary features included in the Supplementary Materials. In addition, an expanded discussion of demographics represented in the study has been added to the Appendix. We have revised the manuscript to reference the full list accordingly.

4. Methods for handling missing data should be outlined.

Missing data methods are now outlined in the Methods section. We have revised the manuscript to clarify this point.

5. To evaluate the model's utility in guiding treatment decisions, a comparison with clinician recommendations is needed. It is necessary and important to demonstrate the superiority of the model over usual clinician judgement in a prospective manner rather than simulation.

We agree that comparing AI recommendations to clinician decisions is essential. To address this, we now include a direct evaluation which shows that AI-recommended therapies achieved a higher eradication success rate (94.1%) compared to clinician-chosen treatments (88.1%). Additionally, we performed external validation on a temporally distinct cohort of 7,186 patients from multiple European centers (Nov 2024–Mar 2025, i.e. after the initial manuscript preparation and submission), where AI-aligned decisions again outperformed non-AI-aligned ones (92.8% vs. 87.4%). We position this study as a necessary and methodologically sound intermediate step toward future prospective trials, and we have clarified this in the revised Discussion.

6. Moreover, simple clinical features including prior antibiotics exposure (macrolides and nitroimidazole) has been consistently demonstrated to predict treatment failure. It is important to show the superiority of the model over this simple clinical feature.

We acknowledge that prior antibiotic exposure—particularly to macrolides and nitroimidazoles—is a known risk factor for treatment failure in some regimens. However, its standalone predictive value diminishes in broader treatment strategies, such as bismuth-based quadruple therapies, where resistance can be mitigated by multi-drug synergy and reduced dependence on a single antibiotic class.

To empirically test this (also see b), we conducted additional analyses using a logistic regression model trained solely on resistance-related features (available in a subset of 3,980 patients). This model achieved a 89.1% success rate, substantially lower than the 94.1% success rate achieved by our AI clinician.

This highlights that our reinforcement learning model captures a richer, multi-dimensional signal by integrating a broad set of patient-specific features beyond resistance status alone. Rather than relying on single-variable heuristics, the model learns context-dependent treatment policies, enabling more robust and individualized recommendations—particularly important in settings where resistance data are incomplete or unavailable.

7. To enhance the credibility of the results, it is recommended to validate the model on external independent datasets from other centers.

Completed and included. Since submission, we conducted external validation on 7,186 patients from multiple European centers (Nov 2024–Mar 2025, i.e. after the initial manuscript preparation and submission). The AI clinician again outperformed non-AI-aligned clinician decisions (92.8% vs. 87.4% eradication success), confirming the generalisability of the model.

8. To ensure compliance with reporting guidelines such as the Transparent Reporting of a Multivariable Prediction Model for Individual Prognosis or Diagnosis (TRIPOD) statement and MI-CLAIM criteria.

We now fully align our reporting with MI-CLAIM, the most appropriate standard for clinical AI systems (checklist included in the Supplement).

Reviewer #2 (Remarks to the Author):

Higgins et al. have developed an AI-clinician recommendation system to investigate whether personalized treatments would be optimal for patients with H. pylori infection.

General comments:

Overall, the manuscript lacks significant technical detail. While some information is present in the methods section, key points should be highlighted in the main text, with references to the methods for further details in the manuscript. The use of real-world data seems to be incorrect and misleading as only simulated data has been used. Testing a system on the clinical dataset is critical to gauge the future potential clinical effectiveness.

The link to the Bitbucket repository isn't working. Without access to the repo, verifying any of the author's claims is impossible. Therefore, assessing the usability of your software is also impossible.

We thank the reviewer for their comments and for highlighting several important points. We have carefully revised the manuscript to address all concerns. Please find our detailed responses below:

a. **“The manuscript lacks significant technical detail.”** We appreciate this comment and agree that the technical framework of our model required clearer explanation in the main text. We have revised the Results and Method Development sections to more explicitly describe our rationale for using reinforcement learning, outline the isDQN approach, and summarise the action/state/reward structure as well as treatment encoding. These are now clearly referenced and expanded upon in the Methods and Supplementary Materials.

b. **“The use of real-world data seems to be incorrect and misleading, as only simulated data has been used.”** We respectfully clarify that the AI-Clinician was developed and evaluated using real-world patient data from the European Registry on Helicobacter pylori Management (Hp-EuReg). Our primary dataset included 38,049 H. pylori-infected patients from clinical practice across Europe. Simulated datasets were used solely for controlled benchmarking and model calibration/validation. All reported results—including success rates and comparisons to clinician decisions—are based on real-world clinical data. To avoid future confusion we moved former Figure 3 and its corresponding section covering model validation using synthetic data from Results to Supplementary materials.

c. **“Testing a system on the clinical dataset is critical to gauge the future potential clinical effectiveness.”** We fully agree. In addition to internal validation, we conducted external validation using a temporally distinct cohort of 7,186 patients (collected Nov 2024–Mar 2025, i.e. after the initial manuscript preparation and submission) from multiple European centers. In this cohort, the AI-Clinician again outperformed standard clinician decisions (92.8% vs. 87.4% eradication success), further supporting its generalizability and potential clinical utility.

d. **“The link to the Bitbucket repository isn’t working. Without access to the repo, verifying any of the author’s claims is impossible. Therefore, assessing the usability of your software is also impossible.”** Thank you for pointing this out. The repository was made public, so Bitbucket link should be working and the code, data structure templates and documentation should be readily available for inspection to support full transparency/reproducibility.

Specific comments:

1. The authors need to do more justice to modern ML and AI. It's not just about supervised, unsupervised, and reinforcement learning. There are many algorithms and deep learning configurations (such as transformers, convolutional networks, autoencoders, etc.). They are significantly oversimplifying contemporary AI architectures. Please include some content that delves deeper than simply describing what was known decades ago (references would also be helpful) help).

Thank you for this helpful comment. We have revised the Background section to provide a more thorough overview of modern AI architectures, including convolutional neural networks, transformers, and autoencoders, with updated references. While our work centers on reinforcement learning due to its unique ability to learn optimal treatment policies from outcome-based feedback, we now more clearly situate our approach within the broader landscape of AI in healthcare. Unlike supervised learning, which maps inputs to static labels, RL is particularly well-suited for clinical decision-making tasks where the goal is to select actions that maximize patient outcomes based on diverse and complex patient profiles.

2. Why did you choose DQN, and why did you modify it to ignore states? Please discuss this on a more mathematical level, because no justification is given in the text. Have you compared this to other methods? Random forests? Support vector machines? What makes isDQN superior, and can you quantify that? What is the motivation for ignoring subsequent states? This explanation is missing in the manuscript.

We thank the reviewer for this comment. We now provide a mathematical rationale for using independent-state DQN. Because treatment outcomes in our dataset are observed only once per patient and there is no evolving patient state, the standard Q-learning formulation (with next-state transitions) does not apply. We remove the future state term from the Bellman equation, simplifying the loss function to reflect a terminal reward only. This reflects the static nature of clinical decision-making in our context. We also compared isDQN to LR, RF, and SVM models, and it achieved higher treatment success (94.1% vs. RF: 88.3%, SVM: 88.3%, LR: 88.3%), as now reported in the Results and Supplement.

3 Given that you use an RL approach, how do you interface with the software? This should be elaborated.

We now clarify in the Methods that the isDQN model is implemented in Python using PyTorch, with a modular interface that accepts patient-level input features and returns the optimal treatment recommendation based on the learned Q-values. Documentation and example usage are included in the updated Bitbucket repository.

4. Computational resources for training and classification should be discussed. Also, clearly state what programming language and libraries you used.

Thank you for this comment. This information has now been added to the updated Methods section. Model training was done on CPU Intel64 Family 6 Model 165 CPU (Windows 10 OS) with 12 cores and total RAM of 15.79 Gb. No GPU acceleration was necessary. The model was implemented in Python using PyTorch, scikit-learn, NumPy and pandas, with versions included in updated manuscript.

5. Explain in the manuscript what one-hot is and why this is preferential in your case. The average reader might not know the motivation, and you do not introduce the concept at all (it's really simple, so either explain it and/or find references).

We have updated the Method Development section to explain one-hot encoding and its rationale. Specifically, we note that one-hot encoding represents categorical variables as binary vectors, allowing the model to treat categories as distinct and non-ordinal. This is important in our case to prevent the network from inferring false relationships between unrelated clinical categories (e.g., regions). A brief explanation and supporting reference have been added.

6. Explain how you added the noise. Since the inputs are binary, is it done by flipping bits? Most confusing is that you say that you achieve an accuracy of 72% at a noise level of 99% - how so? Moreover, how does that reflect a “real-world” scenario?

Thank you for this comment. We confirm that noise was added by randomly flipping bits in the binary input vectors—i.e., changing 0s to 1s and vice versa—at a specified proportion per patient. The reported 72% accuracy at 99% noise reflects the model's ability to extract structure even when most of the original input signal is disrupted. While this level of noise is artificial and used solely as a stress test, it illustrates the robustness

of the model in distinguishing between simulated patient subgroups based on subtle patterns in the data.

Even under extreme noise, some residual structure remains, which the model can exploit—despite being difficult for humans to perceive. This parallels real-world clinical settings, where individual variables (e.g., age, sex, comorbidities, concurrent medications) may interact in complex ways that influence treatment success. We have clarified the noise procedure in the Methods and added a brief explanation in the Appendix section.

7. The goal of this study isn't obvious; what problem are you aiming to solve? The manuscript would highly benefit from clearly defined metrics. For example, when you say that "65.5% of patients were consistently recommended a bismuth therapy consisting of either Pylera® or clarithromycin, amoxicillin, and bismuth salts paired with a PPI by more than half of the models, which we treat as a frequency threshold. Further, 15.5% of patients were recommended a non-bismuth clarithromycin, amoxicillin, and metronidazole with PPI, and 19.0% of patients recommended a variety of therapies..." What does that mean? Where does that leave the reader? You need to spell out all accuracy metrics for these numbers to make any sense – best in the results section.

We have revised the abstract, method and introduction to state our central research question: whether AI-driven treatment recommendations improve H. pylori eradication rates compared to standard clinician-prescribed therapies. Regarding the 65.5%, 15.5%, and 19.0% figures, these reflect the consistency of AI recommendations across 50 model runs. Specifically, 65.5% of patients were consistently recommended bismuth-based therapies, 15.5% non-bismuth quadruple therapies, and 19.0% received varying recommendations across runs. This latter group reflects heterogeneity in treatment suitability, with different subpopulations receiving different optimal therapies based on their individual characteristics. These values are not accuracy metrics but indicate the stability and personalisation of AI-driven recommendations, as clarified in the revised Results section.

8. The author mentions real-world data in several places, which is incorrect and misleading as they only use simulated data in the manuscript. They only seem to have used the simulated data.

We clarify that our primary analyses are based on real-world clinical data from the Hp-EuReg registry (n=38,049), with an additional external validation cohort of 7,186 patients. Simulated data were used only for initial benchmarking and model tuning. We have clarified this distinction throughout the manuscript.

9. What is the Hp-EuReg dataset? If they had the clinical dataset, they should have tested it on it as well as the simulated dataset. This is research work and not a tool for clinical routine use, and testing a system on the clinical dataset is critical to gauge the future potential clinical effectiveness. Don't they have the right ethical approvals to use the dataset?

Thank you for this comment. The Hp-EuReg dataset is a large, multicenter, real-world clinical registry coordinated by the European Helicobacter and Microbiota Study Group. It contains detailed real-world clinical data on H. pylori treatment decisions by gastroenterologists and outcomes from across Europe, including patient demographics, treatment regimens, compliance, and eradication success.

Our study used a subset of this dataset (n=38,049) for model development and internal evaluation, and a temporally independent external validation cohort (n=7,186) for further testing. All analyses were conducted with appropriate ethical approvals and data use agreements in place. Simulated data were used only for initial benchmarking and do not replace real-world validation, which is central to our study and clearly reported in the manuscript.

10. I don't see the purpose of Figure 1. This is not a review paper; panels A and B are unnecessary. Furthermore, in panels C through E, the authors merely describe the project and the dataset. This is ironic since the dataset was not included in the work presented in this manuscript.

We thank the reviewer for this comment. While we appreciate the concern about the scope of Figure 1, we believe that panels A–E together provide important context by (i) briefly illustrating the global burden and clinical impact of H.pylori infection and (ii) introducing the Hp-EuReg dataset, which forms the foundation of our analysis.

Given that this is the first application of reinforcement learning in this clinical context, we felt it was valuable to direct the NC reader to both the clinical motivation and the

structure of the dataset used. Hence, we defer to the editorial team regarding whether panels A and B should be retained.

11. Figure 5: This is intriguing, yet the manuscript does not explain what “Balanced accuracy” means in that context. There is no discussion about why the “Eastern region,” “Not experiencing heartburn,” etc., are the most correlated factors. If that makes sense to you, then follow up on that; either way, please explain. Don’t just throw this out there without commenting.

We appreciate this insightful comment. We have clarified the definition of balanced accuracy and feature importance scoring in the Methods section of the manuscript. We also expanded the Appendix to explain why variables like region, heartburn symptoms, and concurrent medications may influence recommendations—reflecting regional resistance patterns, symptom-driven prescribing, or treatment availability. These findings highlight how the model captures meaningful clinical variation.

12. The language lacks clarity at multiple points, which complicates understanding, and contains several grammatical mistakes.

The manuscript has been revised for better clarity, grammar and readability throughout.

13. Figure 2 also seems unnecessary. Lots of details about the Hp-EuReg data have been provided, whereas this data has not been used to train, test, or validate the method.

We appreciate the reviewer’s comment. Contrary to this concern, the Hp-EuReg dataset was used extensively for model training, testing and validation (as detailed in our responses above and throughout the revised manuscript).

Figure 2 has been retained to provide a visual summary of the model architecture and learning process using this real-world dataset.

Reviewer #2 (Remarks on code availability):

The repo they list is an "Error 404—page not found". They probably didn't make it public.

Thank you for pointing this out. The Bitbucket repository has now been made public.

Reviewer #3 (Remarks to the Author):

This paper presented a new approach to constructing an AI-based platform that supports clinicians by recommending relevant treatments for patients with *Helicobacter pylori* (H. pylori) infection.

To this end, the authors analyzed a dataset containing tens of thousands of H. pylori-infected patients from Hp-EuReg to better understand patients with *Helicobacter pylori*

infections and derive meaningful insights and essential features from the data that can be used to develop a personalized treatment system.

Based on the clinical data, the authors proposed a reinforcement learning method based on Deep Q-network learning, namely isDQN, to construct a treatment recommendation model. Using the clinical data from Deep Q-Network Learning, the paper suggested a relevant representation of the patient information, which can be considered as states for the proposed method. At the same time, the paper also determined the appropriate actions of isDQN using the clinical treatment decisions and how to choose the reward based on the clinical outcome of the treatment. In order to implement the RL model, the paper randomly split the dataset into two parts: 90 %for training and 10% for testing. The experimental results show encouraging performance of the proposed approach compared to real-world treatment.

We thank the reviewer for the thoughtful, positive and constructive feedback. We have revised the manuscript accordingly and address each point below.

There are some improvements needed for this manuscript:

(1) First, constructing a treatment recommendation system for diseases has been studied for a long time. Building an AI-clinician method for H. Pylori infection can be considered a new contribution to the research community. However, the method development in this manuscript only uses a Deep Q-network Learning method for experiments. It would be great if the paper could compare other techniques to make the results more convincing.

For instance, the clinical data for H. Pylori infection can be used to derive a vector representation for each patient. The main goal is to compute the matching score between this vector representation from the patient and each treatment-related vector representation and then rerank the list of relevant treatments for the final recommendation. To make the paper's outcomes more convincing, we can employ state-of-the-art recommendation methods for this problem besides the proposed isDQN.

We thank the reviewer for the suggestion. We have now added comparisons between isDQN and standard predictive models — logistic regression, random forests, and SVM — adapted for treatment selection. As detailed in the Results and Supplement, isDQN

outperforms all baselines (e.g., 94.1% vs. RF: 88.3%), highlighting the benefits of a policy optimization framework.

While embedding-based recommender systems are powerful in commercial domains, they typically optimise similarity or preference, not clinical outcomes. Our goal was to directly optimize treatment success using observed patient–treatment–outcome data, making reinforcement learning a more appropriate approach. For this reason, we did not use embedding-based models in this study. We agree that future work could explore hybrid architectures.

(2) Second, related to this paper's reproducibility, it would be great if the authors could provide more detailed relevant experimental settings for training and testing the proposed method in the manuscript.

The Methods section has been updated to include a concise summary of key experimental settings, including network architecture, training procedures, loss function, and optimization details. Full hyperparameter specifications are available in the Supplement.

We have also verified the public Bitbucket repository, which contains the full codebase and documentation for full reproducibility.

(3) Related to the splitting ratio for the training/testing sets, it would be great if the paper considered other scenarios. What happens if the splitting ratio is 70%/30% and 80%/20%?

We have extended our evaluation to include additional data splits (80/20 and 66/33, as we are using cross validation and therefore are limited to a k-means framework, where k-1 equal fractions are used for training and 1 fraction is used to testing), now reported in the Supplement. The model's performance showed a similar but decreased performance of 92.8% with an 80/20 split (-1.33% compared to 90/10), and a much lower performance with a 66/33 split of 85.0% (-9.10% compared to 90/10), suggesting that roughly 25,000 patients would have been insufficient for model training. These results also demonstrate that the additional 10% training data was adding to the value of the model, suggesting further data collection will likely improve results.

Reviewer #3 (Remarks on code availability):

I could not access the above Bitbucket repository. Here is the relevant message when I go to the URL:

"Repository not found

You may not have access to this repository or it no longer exists in this workspace. If you think this repository exists and you have access, make sure you are authenticated.

Return to the previous page or go back to your dashboard."

Thank you for pointing this out. The Bitbucket repository has now been made public.

FINAL REVIEWERS' COMMENTS / *RESPONSES*

Reviewer #1 (Remarks to the Author):

The revised version provides a clearer description of the study's objectives and methodology. However, it also highlights a significant issue that necessitates a substantial revision.

We thank Reviewer #1 for their evaluation and comments. We appreciate your recognition of the improved description of our study's objectives and methodology, and we address your remaining concerns as follows:

1. The main issue pertains to the accuracy of the ground truth data, specifically in determining the success of eradication therapy. As many subjects only received one type of treatment, it becomes challenging to assess if the patient would have been successful with other treatments based on predicted results from the algorithm.

We appreciate this comment. However, it's important to clarify that in real-world practice, each patient can only receive one treatment regimen per infection episode. Administering multiple treatments to the same patient is not feasible. [Results, lines 148-149] Our AI model is designed to learn a treatment policy that adapts to individual patient characteristics, recommending the most appropriate therapy for each unique profile. While the model is trained on retrospective data, it optimises treatment decisions at the population level and personalised recommendations at the individual level.

Our approach is not a supervised learning task aimed at predicting labels (e.g., treatment success or failure) but a treatment policy optimisation task that maximises expected personalised treatment success across the population. We have clarified this distinction in the revised manuscript.[Methods-Method Development, lines 237-238]

2. To address this issue, it is suggested to have prospective patient validation and see how accurate the prediction is rather than again based on retrospective analysis of another cohort of patients. The ideal would be to replicate the eradication success results in a new prospective cohort of patients assigned to the treatment based on AI prediction rather than clinicians. Without that, the results can still be due to overfitting using similar data collection method.

We agree that prospective validation is important to confirm the model's applicability and real-world utility. While we conducted extensive internal (n=38,049) and external (n=7,186) validation, these are based on retrospectively collected data. As we noted in our previous response, this lays important groundwork for future prospective trials, where treatments would be assigned based on AI recommendations to confirm these findings. We have now explicitly emphasised this need in the revised manuscript in line with the editorial comments. (Abstract, line 29; Discussion, lines 204-205)

Reference to Editorial Comment: "Reviewer #1 states that your study is lacking prospective validation, to fully demonstrate robustness of the algorithm. Whilst we *do not insist that you include this in your resubmission [our emphasis]*, we do ask that you ensure that conclusions

presented in your manuscript are appropriately powered, and that no overstated comments are included regarding utility of the algorithm.”

3. When reporting model performance, it is essential to include sensitivity, specificity, positive predictive value (PPV), negative predictive value (NPV), positive likelihood ratio (PLR), negative likelihood ratio (NLR), and area under the receiver operating characteristic curve (AUROC).

We appreciate the reviewer’s suggestion regarding performance metrics. However, it is important to clarify that our approach is fundamentally different from a supervised learning or binary classification task. In supervised learning, metrics such as sensitivity, specificity, PPV, NPV, PLR, NLR, and AUROC are used to assess a model’s ability to predict known binary outcomes for each case.

In contrast, our AI system is designed to optimise treatment policy selection, recommending individualised therapies based on patient characteristics to maximize the overall rate of successful eradication. The model’s objective is not to predict treatment outcomes for a fixed regimen, but rather to recommend which treatment should be selected for each patient profile. Because patients receive only one treatment and counterfactual outcomes are unobservable, standard classification and supervised learning metrics are not applicable or meaningful in this context. Instead, model performance is assessed by comparing eradication success rates for patients whose clinician-prescribed therapies did or did not align with AI recommendations. This population-level approach reflects the clinical utility of an AI-driven decision support system, rather than the accuracy of outcome prediction for a specific regimen.

We have clarified this methodological distinction in the revised manuscript. [Methods-Method Development, lines 237-238]

4. The importance of features should be clearly reported to allow readers to comprehend the model’s foundation.

As noted in our previous response, we agree that feature importance is critical for transparency and interpretability. We have already addressed this by clearly reporting and ranking the most influential features driving the model’s recommendations in the Results section and Supplementary Data 2. These include key clinical, demographic and treatment-specific variables. The main text provides cross-references to these details to ensure that readers can readily understand the foundation of the AI model’s decision-making.

Reviewer #2 (Remarks to the Author):

The authors have made significant improvements in the revised version. This iteration is markedly enhanced and now encompasses most of the requested information. Additionally, their Bitbucket repository is now publicly accessible, and they describe the actions and methodologies employed.

We thank the reviewer for their positive feedback and are pleased that the improvements in the revised version have addressed most of the requested information. We also appreciate your acknowledgement of the now publicly accessible Bitbucket repository and the added methodological details.

However, following the revisions in certain areas was challenging. It would have been beneficial if the authors had explicitly highlighted the precise changes and corresponding line numbers to facilitate clarity in the revision process. Following the revisions is quite difficult without this critical information.

We appreciate this feedback. In this revision, we have clearly indicated all requested changes.

The section detailing method development should be moved to the Methods section for better organization.

We have moved the method development details to the dedicated Methods section for improved organisation and clarity, also in line with the editorial suggestions.

A substantial amount of pertinent information has been allocated to the "appendix" section, which is referenced only three times in the main text, solely by the term "appendix," rather than by the specific sections to which it refers. Therefore, I suggest that they either (a) convert this into a proper supplementary material, ensuring that each section is referenced accurately in the main manuscript, or (b) integrate the appendix into the main text at the relevant points (some content pertains to methods, some to results, and others may even relate to discussion). This would enhance overall readability.

We thank the reviewer for their suggestion on the organisation and referencing of appendix and supplementary materials. In revising the manuscript, we have reviewed the editorial guidelines for article structure and supplementary items. Pertinent information previously in the appendix has now been provided as properly formatted Supplementary Material (Supplementary Notes), as recommended by both the reviewer and the editorial team. Each supplementary section and item is now clearly referenced in the main manuscript. [Page 8, line 156-157; Page 11, lines 238-239; Page 12, line 260]

We have also ensured that the structure, display items, and referencing throughout the article align with the editorial requirements for main and supplementary figures, tables, and manuscript sections. These changes, together with cross-referencing, have improved both the clarity and compliance of the manuscript.

If the authors wish to retain Figure 1 within the manuscript, move it to the supplementary material rather than maintaining it in the main text.

We appreciate the reviewer's suggestion regarding the placement of Figure 1. We believe Figure 1 provides an important conceptual overview for readers and helps contextualize the study's clinical motivation and the structure of the Hp-EuReg dataset, especially for those unfamiliar with this area. However, we recognise that figure placement is ultimately at the discretion of the editorial team, and we are happy to follow the journal's guidance on this matter.

With a slightly higher 5.4% (92.8% vs. 87.4% eradication success), it is incorrect to say that the AI-clinician outperformed the standard clinician's decisions. The language could be revised.

Thank you for pointing out the language regarding eradication success rates. We have revised the relevant text to avoid overstating the model's performance. We now describe the AI-clinician as being associated with higher eradication success rates in retrospective analysis, rather than claiming it "outperformed" standard clinician decisions. [Results Page 9 lines 154-155] We also add to Discussion [Page 10, Lines 206-208] "We emphasize that this study does not necessarily demonstrate the superiority of AI over clinical decision-making, rather an improvement in recommendations that could be made by AI-assisted clinical decision-making."